# Liver Transplantation for Hepatocellular Carcinoma: A Real-Life Comparison of Milan Criteria and AFP Model

**DOI:** 10.3390/cancers13102480

**Published:** 2021-05-19

**Authors:** Bleuenn Brusset, Jerome Dumortier, Daniel Cherqui, Georges-Philippe Pageaux, Emmanuel Boleslawski, Ludivine Chapron, Jean-Louis Quesada, Sylvie Radenne, Didier Samuel, Francis Navarro, Sebastien Dharancy, Thomas Decaens

**Affiliations:** 1Faculty of Medicine, University Grenoble-Alpes, 38000 Grenoble, France; bbrusset@chu-grenoble.fr; 2CHU Grenoble-Alpes, 38000 Grenoble, France; lchapron@chu-grenoble.fr (L.C.); jlquesada@chu-grenoble.fr (J.-L.Q.); 3Hospices Civiles de Lyon, Hôpital Edouard Herriot, 69003 Lyon, France; jerome.dumortier@chu-lyon.fr; 4Assistance Publique des Hôpitaux de Paris, Hôpital Paul Brousse, Centre Hépato-Biliaire, 94800 Villejuif, France; daniel.cherqui@pbr.aphp.fr (D.C.); didier.samuel@pbr.aphp.fr (D.S.); 5CHU de Montpellier, 34295 Montpellier, France; gp-pageaux@chu-montpellier.fr (G.-P.P.); f-navarro@chu-montpellier.fr (F.N.); 6CHU de Lille, 59000 Lille, France; emmanuel.boleslawski@chru-lille.fr (E.B.); sebastien.dharancy@chru-lille.fr (S.D.); 7Hospices Civiles de Lyon, Hôpital de la Croix Rousse, 69004 Lyon, France; sylvie.radenne@chu-lyon.fr; 8Institute for Advanced Biosciences, Research Center UGA/Inserm U 1209/CNRS 5309, 38000 Grenoble, France

**Keywords:** AFP model, Milan criteria, liver transplantation, hepatocellular carcinoma, post-transplant recurrence

## Abstract

**Simple Summary:**

The α-fetoprotein (AFP) model officially replaced the Milan criteria in France for liver transplantation (LT) for hepatocellular carcinoma (HCC) in January 2013. The aim of our retrospective study was to analyze the agreement of the criteria and the results of LT with an intention-to-treat design since the adoption of the AFP model and to compare them to the practice and results of LT before the adoption of the AFP model. We did not observe significant changes in practices in 523 consecutively listed patients, with a good agreement (88%) to AFP criteria on the explants before and after the adoption of the AFP model. However, the prognosis of patients listed in the most recent period was worse, maybe because of a significant increase in bridging treatments and in the waiting time. This observational study provides an insight into the real-life course of LT for HCC.

**Abstract:**

Purpose: To compare the agreement for the criteria on the explant and the results of liver transplantation (LT) before and after adoption of the AFP (α-fetoprotein) model. Methods: 523 patients consecutively listed in five French centers were reviewed to compare results of the Milan criteria period (MilanCP, *n* = 199) (before 2013) and the AFP score period (AFPscP, *n* = 324) (after 2013). (NCT03156582). Results: During AFPscP, there was a significantly longer waiting time on the list (12.3 vs. 7.7 months, *p* < 0.001) and higher rate of bridging therapies (84 vs. 75%, *p* = 0.012) compared to the MilanCP. Dropout rate was slightly higher in the AFPscP (31 vs. 24%, *p* = 0.073). No difference was found in the histological AFP score between groups (*p* = 0.838) with a global agreement in 88% of patients. Post-LT recurrence was 9.2% in MilanCP vs. 13.2% in AFPscP (*p* = 0.239) and predictive factors were AFP > 2 on the last imaging, downstaging policy and salvage transplantation. Post-LT survival was similar (83 vs. 87% after 2 years, *p* = 0.100), but after propensity score analysis, the post-listing overall survival (OS) was worse in the AFPscP (HR 1.45, *p* = 0.045). Conclusions: Agreement for the AFP model on explant analysis (≤2) did not significantly change. AFP score > 2 was the major prognostic factor for recurrence. Graft allocation policy has a major impact on prognosis, with a post-listing OS significantly decreased, probably due to the increase in waiting time, increase in bridging therapies, downstaging policy and salvage transplantation.

## 1. Introduction

The success of liver transplantation (LT) for hepatocellular carcinoma (HCC) is ruled by the necessity of similar outcomes for HCC and non-HCC recipients, as they directly compete in a large waiting list, with a system of prioritization that is constantly in question. Indeed, the risk of tumor recurrence has to be the lowest, and HCC candidates for LT should be strictly selected in the context of organ shortage. 

International guidelines have considered the Milan criteria as the standard for selecting HCC patients for deceased donor LT [1] as a guarantee of good outcomes, with an actuarial survival rate of 75% after four years in the original publication in 1996 [2], confirmed in 2011 [3]. ‘Expanded’ criteria, developed to allow a wider access for patients with HCC that may receive clinical benefit from LT, failed to enter the last European and American recommendations [4,5].

On the other hand, the prognostic information of the AFP value at the time of listing, and probably even more at the time of LT, has proved its worth over time and is now well established [1]. It has been shown that the kinetic of AFP, with an increase of more than 7.5 ng/mL/month, is a predictor of post-LT recurrence [6], while its response to locoregional therapies is a predictor of good outcomes [7]. A rate of AFP > 1000 ng/mL could represent an exclusion criterion for LT [8], as proposed by the UNOS in 2016. Combined with the response to locoregional therapies (LRTs), the AFP response identifies good transplant candidates [9].

The AFP model (Table 1) established in 2012 [10] accounts for AFP level, number of nodules and the size of the largest nodule. It has been proven to outperform the Milan criteria in identifying candidates with low risk of HCC recurrence or who will survive for 5 years after LT, as per the revised Up-to-Seven criteria, also called Metroticket V2.0 model [11], developed after the Up-to-Seven criteria [12]. The strength of the AFP model lies in the demonstration of classification improvement between low- and high-risk HCC. Because Milan criteria were regularly overstepped, the AFP model has been endorsed in France since January 2013, and HCC candidates must now have an AFP score ≤ 2 to remain or be reintegrated within the waiting list after downstaging.

Since 2007, the French allocation system has been based, except for emergency transplantation (acute liver failure, primary nonfunction), on a common score called ‘liver score’, which essentially considers the MELD score and the time spent on the waiting list, with the burden for this latter parameter depending on the indication of transplantation. HCC patients obtain a higher ‘liver score’ and get access to LT more quickly than patients with isolated cirrhosis, and compete with them after 14 to 18 months even when they have a low MELD score. It is worth noting that another important change in the French practices occurred around the year 2012, depending on centers, with the generalized use of ‘temporary contraindications’ (TCIs) among patients for LT, which basically means ‘temporary inactivation’ on the waiting list without losing the allocation points gained by the time spent on the waiting list. Patients with HCC controlled after a curative treatment or exceeding allocation criteria can be ‘temporarily contraindicated’ until reassessment of the LT indication (recurrence or successful downstaging).

In this French multicentric retrospective study, we first aimed at comparing the agreement for the AFP score on the explant analysis before and after the adoption of these criteria. The secondary goal is to compare the general results of LT in terms of tumor recurrence, dropout rate, overall survival and disease-free survival before and after the AFP model implementation.

We aimed at examining how the AFP model performed in prioritizing patients who would have been excluded by Milan, but who had acceptable outcomes, and conversely, whether it identified patients within Milan criteria but who would have had inferior outcomes.

Finally, a large cohort and a large volume of data give us the opportunity to discuss the impact of downstaging, waiting time and response to bridging therapies.

## 2. Patients and Methods

### 2.1. Study Design

All patients registered consecutively for LT because of an HCC between March 2011 and March 2014 on the ABM (Agence de Biomédecine, French Agency for organ sharing) listing in five French centers were included, whether they finally underwent LT or not, in order to analyze the overall results of LT in an intention-to-treat design. The centers were Centre Hepatobiliaire Paul Brousse (Villejuif), Montpellier, Lille, Lyon and Grenoble. Data cutoff was April 2017. The entire French database could not be analyzed because of the lack of accessibility of data.

### 2.2. Patients

A total of 557 consecutive patients were screened to participate in this study. The patients whose explants did not reveal HCC or for whom HCC was not the first indication for listing were excluded. We included living donors (*n* = 9), domino grafts (*n* = 11), partial transplantation (*n* = 9), exceptional graft (*n* = 1) or expert component (*n* = 2) in order to keep the studied population as close to real patients in the daily practice as possible. For the same reason, patients who had spent more than one year or were still listed in ‘temporary contraindication’ because of the absence of tumoral progression were maintained in the analysis.

The study was approved by the Ethics Committee (CECIC of Auvergne Rhone Alpes, IRB file 2015-31) and authorized by CCTIRS and CNIL Committee regarding the use of patient data (NCT 03156582).

### 2.3. Data Collection

Data collection was retrospectively done on each site. Pretransplant data included demographics, cause of cirrhosis, histological data at time of diagnosis, imaging tumor features at first diagnosis, listing as a salvage transplantation (defined as the presence of a curative treatment done more than one year before listing) and AFP values at the time of diagnosis.

The official registered number and size of tumors and the AFP values declared to ABM were prospectively collected, as were the imaging tumor features collected at the time of listing.

Pre-LT bridging therapies and their results, the last imaging tumor features and the last AFP values (within 3 months before LT) were retrospectively collected. The choice of bridging therapies could slightly differ according to the radiological or surgical skills of each local tumor board meeting, but the indications were still the existence of ‘active’ nodules because of viable tumor tissue on imaging. Response to treatment after locoregional therapy was assessed according to the mRECIST criteria, considering the size and number of the residual viable tumor tissue. Downstaging policy was defined as a reduction in the size of tumor using locoregional therapies, when the patient was outside the criteria in use at the time of the imaging, without upper limit restriction such as UCSF criteria. The success of downstaging was defined by the reintegration to the Milan criteria, whatever the period, to be homogeneous.

The pathological features of HCCs were collected after LT from the explant reports.

Post-LT follow-up data included death, cause of death, HCC recurrence, date of tumor recurrence and date of last follow-up visit. The diagnosis of tumor recurrence was established based on imaging reports, histological reports and/or multi-disciplinary tumor board reports. In the intent-to-treat design of our study, cholangiocarcinoma diagnosis at the time of tumor recurrence was included. All those seven patients had typical HCC on pre-LT imaging and cholangiocarcinoma component on the explant analysis.

### 2.4. Statistical Analysis

Data were expressed as median and interquartile range. The chi-squared test was used for categorical variables, or a Fisher test was used for small samples. A nonparametric test (Mann–Whitney U) was used for numerical variables. Patient survival rates were estimated first with the Kaplan–Meier method and compared with the log-rank test, and a Fine and Gray model was used to take into account the competitive risk due to death. Competitive risk analysis was used to analyze the probability of tumor recurrence. Univariate and multivariate analyses were performed as exploratory analyses, and only on variables with clinical significance for the outcomes of post-transplant survival and recurrence of HCC after LT. Variables with *p* < 0.15 in univariate analysis were tested in the multivariate Cox proportional hazard model to identify independent prognostic factors. Because of differences in baseline characteristics between the two groups, a propensity score analysis was used. The propensity score has been established on 17 parameters chosen for their clinical pertinence at diagnosis or listing: age, sex, cirrhosis, the presence and number of curative treatments realized before listing to avoid transplantation, the time between diagnosis and listing, the MELD score at listing, the number of tumors, largest diameter and total diameter as well as the AFP value at listing, the noncompliance of the AFP model and of the Milan criteria at listing, the presence and number of bridging therapies while awaiting for LT, the median waiting time for LT, and a downstaging policy. The matching method on this propensity score was done to evaluate the main criteria and compare survival rates.

An independent statistician performed the statistical analyses. Stata software, version 14.2 (StataCorp, College Station, TX, USA), was used for statistical analysis at the Centre d’Investigation Clinique Plurithématique of Centre Hospitalier Universitaire Grenoble Alpes.

## 3. Results

After exclusions, the final study population consisted of 523 patients (Figure 1). A total of 364 patients were given a transplant, whereas 159 patients either dropped out (*n* = 146) or did not undergo a transplant because of complete tumor response at data cutoff (*n* = 13).

The group of patients of the MilanCP was composed of patients who were either given a transplant or dropped out of the list at the time of the Milan criteria use (up to 1 June 2013 because of a mandatory re-evaluation of patients by the various teams in order to conform to the criteria on this date). The group included only 199 patients.

The group of patients of the AFP score period (AFPscP) was composed of patients who underwent liver transplantation or dropped out of the list after 1 June 2013, i.e., after the implementation of the AFP score by the ABM and reassessment of patients in each center. This group included 324 patients.

### 3.1. Patients’ Characteristics

Patients’ characteristics are summarized in Table 2. Initial tumor characteristics were similar in both groups. At listing, about 25% of patients were listed in a salvage transplantation policy; AFP score was ≤2 in 97.5% of the cases during Milan criteria period (MilanCP) compared to 94.8% of the cases during AFP score period (AFPscP) (ns). The proportion of patients with advanced cirrhosis (Child C, MELD > 20) was 13.6% in the MilanCP vs. 7.7% during the AFPscP.

During the waiting time, 75.4% of MilanCP patients received bridging treatment compared to 84.3% during the AFPscP (*p* = 0.012), and the number of treatment procedures was significantly higher during the latest period of time (*p* < 0.001). In the MilanCP, 274 bridging treatments were realized: complete response (CR) was obtained after 97 treatments (35.4%), partial response (PR) after 119 treatments (43.4%), stable disease (SD) after 17 treatments (6.2%) and progressive disease (PD) after 27 treatments (9.8%). In the AFPscP, 621 bridging treatments were realized: CR was obtained after 219 treatments (35.3%), PR after 266 treatments (42.8%), SD after 43 treatments (6.9%) and PD after 73 treatments (11.7%). The proportions were not different between the two periods (*p* = 0.633).

The rate of downstaging remained around 34% of patients, which was similar in the two periods (*p* = 0.771), and its success rate, around 50%, was also comparable (*p* = 0.829). Among the AFPscP patients, 57% had been placed during their waiting time in ‘TCI’ (temporary contraindication), with a median time of 120 days, vs. only 34% of MilanCP patients, with a median time of 66 days (*p* < 0.01). The reasons were a complete tumor response to a waiting therapy, or progression requiring additional treatment, or alternative causes needing a reassessment of the patient for maintaining them on the list (mainly alternative cancer or alcohol relapse).

In both groups, last imaging occurred within three months before LT. No difference was found either in the number and size of nodules or in the AFP value between groups on the last imaging.

Median waiting time increased from 7.7 months to 12.3 months (*p* < 0.001). Excluding patients still on the list at the time of data cut-off (12 patients during AFPscP), the success rate of liver transplantation was significantly lower during AFPscP (69.1%) compared to MilanCP (76.4%; *p* = 0.008). Around 8.2% of patients died of postoperative complications, rejection or sepsis in the weeks after LT, without any differences between subgroups.

### 3.2. Agreement for the Allocation Criteria on Explant Findings

By agreement to criteria, we mean that the number and size of tumors on the explants, coupled with the last alpha-fetoprotein rate, were in accordance with the criteria. The agreement for the histological AFP score was good and similar in both groups: 87.5% in MilanCP and 88.2% in AFPscP (*p* = 0.838) (Table 2). No difference was found in the number, size, tumor differentiation and other pejorative histological criteria between the two groups. The propensity score analysis confirmed the absence of significant difference in the agreement for the AFP model on explants after adjustment for baseline characteristics with a *p*-value of 0.449.

Risk factors for being outside the AFP model on explant analysis were assessed (Table 3). On univariate analysis, being outside Milan criteria on the last imaging, having an AFP score exceeding 2 on last imaging, a high number of bridging therapies and a downstaging policy were significantly associated with being outside AFP score on the explant. Analysis by a multivariate logistic regression model identified only two independent predictors: last-evaluation AFP score exceeding 2 and a downstaging policy. No influence of the transplant period was found.

### 3.3. Post-LT Tumor Recurrence

Forty-two patients presented a tumor recurrence during the follow-up: 14 (9.2%) in MilanCP and 28 (13.2%) in AFPscP (*p* = 0.239), with an obvious difference in follow-up length between the groups. On these tumor recurrences, seven were cholangiocarcinoma on tumor analysis, but a minority of the recurrences had histology confirmation.

On last-imaging assessment, recurrence rate was not different for patients fulfilling AFP score criteria, whether or not they exceeded Milan criteria, whatever the period (Table 4). Among the few patients transplanted despite exceeding the AFP score criteria, recurrence occurred in 37.5% of the cases during MilanCP and in 50% of cases during AFPscP.

Based on explant findings, recurrence rate was only 4 and 5% depending on the period when patients were transplanted within Milan criteria and AFPsc criteria (Table 4), but it was significantly higher for patients fulfilling AFP score but beyond Milan during AFPscP (22%) compared to MilanCP (11.8%; *p* < 0.05). Same results were observed for patients exceeding AFP score and Milan criteria with significantly higher recurrence during AFPscP (48%) compared to MilanCP (33%; *p* < 0.05).

Risk factors for tumor recurrence were assessed (Table 5). The univariate analysis showed a strong correlation of recurrence with expected histological pejorative criteria, but also with a salvage transplantation procedure (SHR 2.01, 95%CI (1.09–3.71), *p* = 0.025), and with a bridging procedure, especially if there was a downstaging policy (SHR 3.54, 95%CI (1.94–6.48), *p* < 0.001). Moreover, using the threshold of 14.5 months corresponding to the last quartile of our population, a long waiting time on the list was associated with recurrence (SHR 2.02, 95%CI (1.06–3.84), *p* = 0.032). The histological AFP score was a strong predictor (SHR 6.85, 95%CI (3.69–12.71), *p* = 0.001), as was the cholangiocarcinoma component (SHR 6.98, 95%CI (3.46–14.10), *p* < 0.001).

The multivariate analysis showed that the AFP score period was associated with tumor recurrence, independently of six other factors: salvage transplantation policy, downstaging policy, high-risk AFP score on last imaging and on the explants, microvascular invasion and cholangiocarcinoma component on the explants.

However, after propensity score matching on baseline characteristics, 2-year recurrence-free survivals were 92.1% (95%CI (86.1–95.5)) in MilanCP vs. 85.8% (95%CI (79.8–90.1)) in AFPscP (HR 1.38 (0.64–3.00), *p* = 0.412), as assessed by Cox matched on the propensity score estimates (Figure 2a).

Risk of tumor recurrence assessed by competing risk analysis, considering the competing risk of non-HCC-related death, was estimated by the Fine and Gray model and showed similar results, with a subhazard ratio of 1.30 (95%CI (0.43–3.98)), *p* = 0.645, and thus finally no higher risk in the AFPscP. Cumulative incidence of tumor recurrence after 3 years was 5.8% in AFPscP vs. 4.3% in MilanCP (Figure 2b).

### 3.4. Overall Survival

Post-transplant survival, without or with propensity score analysis, was similar in the two groups (Figure 3a). Two-year post-transplantation survival was 87.4% (95%CI (80.9–91.8)) during Milan criteria period vs. 82.7 % (95%CI (76.6–87.4)) during the AFP score period (*p* = 0.100).

The multivariate analysis identified only the histological AFP score beyond 2 as a significant risk factor of death (HR 3.84, 95%CI (1.52–9.74), *p* = 0.005), even if belonging to the AFP period tended to be a pejorative factor (HR 2.86, 95%CI (1.00–8.18), *p* = 0.051) (Table 6).

Post-listing survival assessed by Kaplan–Meier was similar in the two groups, with a three-year post-listing survival rate of 68.2% (95%CI (61.2–74.2)) during MilanCP compared to 66.7% (95%CI (61.2–71.6)) during AFPscP, *p* = 0.447 (Figure 3b). However, after matching on the propensity score, there was a higher risk of death during AFPscP with an HR of 1.45 (1.01; 2.08), *p* = 0.045. Three-year post-listing survival rate was significantly lower with only 58.7% (50.5; 66) during AFPscP compared to 68.8% (61.1; 75.4) during MilanCP (*p* = 0.045) (Figure 3c).

### 3.5. Downstaging Policy

Among 54 patients included in a downstaging policy in the Milan criteria period, 26 (48%) were successfully downstaged into Milan criteria and were given transplants, and 10 (18%) were given transplants despite a failure of downstaging according to Milan. Out of 26 patients successfully downstaged, 4 patients had a tumor recurrence (15.4%) (Figure 4), without significant difference from the global population (OR 0.96, 95%CI (−0.065–0.19), *p* = 0. 337). Out of 10 patients who underwent a transplant despite failure to downstage, 3 had a recurrence (30%). Despite a trend of higher tumor recurrence (30% versus 15.4%), there was no significant difference after unsuccessful downstaging (OR 0.98, 95%CI (−0.016–0.45), *p* = 0. 335), but it was higher than in the global population (30% vs. 9.2%, OR 2.10, 95%CI (0.12–0.40), *p* = 0.038). Regarding the explant, there was a trend of more complete pathological response if there was not a downstaging policy (20.5% vs. 8.3%, OR 2.81, 95%CI (0.73–16.0), *p* = 0.118).

Among 92 patients included in a downstaging policy in the AFP score period, 44 (47.8%) successfully downstaged into Milan criteria and were given transplants, and only 7 patients (7.6%) were subjected to transplants despite a failure of downstaging. Out of 44 patients successfully downstaged, 10 patients had a recurrence (22.7%), without significant difference from the global population (OR 1.62, 95%CI (−0.021–0.21), *p* = 0.107). Out of seven patients who underwent transplants despite a failure to downstage, four had a recurrence (57.1%). HCC recurrence was significantly higher in the AFPscP subgroup of patients who were given transplants despite unsuccessful downstaging (57.1%) than in the subgroup of patients with successful downstaging (22.7%) (OR 1.93, 95%CI (0.01–0.70), *p* = 0.05) or in the global population of the AFPscP group (13.2%) (OR 3.30, 95%CI (0.18–0.70), *p* = 0.001). Regarding the explant, there was no difference in complete pathological response according to a downstaging policy (14.8% vs. 11.7%, OR 1.3, 95%CI (0.46–4.26), *p* = 0.811).

In the subgroup “downstaging policy”, 3 patients of the 36 transplanted during the MilanCP had a complete pathological response (8.3%), and 6 patients of the 51 transplanted during the AFPscP had a complete pathological response (11.7%), without significant difference (*p* = 0.730).

## 4. Discussion

Our study shows that there was no better agreement for the AFP criteria on the explant since the implementation of the AFP model, with 87.5% during the MilanCP compared to 88.2% during the AFPscP (considering viable tumor only). This result may translate into the fact that the French medical teams did not comply with the Milan criteria (since in the MilanCP period only 65.8% were in the MC on the explant considering viable tumor and only 54.6% considering total volume), but they intuitively anticipated AFP model application. There was a relative improvement in the agreement for the Milan criteria over time (65.8% in MilanCP and 70.8% in AFPscP, *p* = 0.314). This rate is slightly better but similar to the rates reported in the literature, ranging between 47% and 66% [13,14,15,16,17,18].

As expected, the AFP score > 2 on explant was predictive of tumor recurrence (in univariate and in multivariate analysis) and was the unique predictor of post-transplant death. Moreover, our analysis of risk factors of tumor recurrence showed that exceeding the AFP score was dynamic. Indeed, exceeding AFP score at diagnosis or at listing was not predictive of tumor recurrence but the last AFP score at imaging and that at the explant analysis were predictive. This pinpoints the major role of other factors during HCC management, such as salvage LT procedure and downstaging policy, which were associated with tumor recurrence in multivariate analysis.

Despite a similar post-LT overall survival during the two periods of time, a worse prognosis was observed for patients listed during AFPscP, with a significantly lower 3-year post-listing OS after propensity matching for AFPscP (58.7% (50.5; 66)) compared to MilanCP (68.8% (61.1; 75.4); *p* = 0.045). This could be explained not only by a significant increase in patients with bridging treatment and a significant number of treatment procedures during the waiting time but also by a significant increase in the waiting time, significantly longer time spent on temporary contraindication and a nonsignificant increase in dropout during AFPscP.

Focusing on downstaging policy, our study shows that being part of a downstaging policy was an independent predictor for being outside of AFP score on the explant analysis (OR = 5.13 (95%CI 2.45–10.8). A successful downstaging to Milan criteria offers a reasonable tumor recurrence rate while superior, concordant with the results of Yao et al. [19] and many others [20,21,22], but unsuccessful downstaging procedure (which should have been a transplant contraindication) was associated with an unacceptable tumor recurrence rate of 57% in the AFPscP. We retain as a message the caution towards downstaging which remains submitted to MC, since results are slightly worse even within MC, and expanded criteria expose patients to higher rates of recurrence [23]. Again, our data suggest worse results in the AFPscP than in the MilanCP in this analysis. In an interesting analysis on downstaging in the United States, Kardashian et al. [22] found that non-downstaged HCC patients receiving LRT had an independently increased rate of HCC recurrence compared with non-downstaged patients not receiving LRT. The hypothesis that LRT may negatively impact HCC outcomes in poor-biology tumors had been raised, mainly because of ischemic damages in the hepatic tumor microenvironment. It has been shown that the production of COX-2 promotes the epithelial-to-mesenchymal transition (EMT) process and enhances HCC invasion and metastasis [24]. The higher number of LRTs in the AFPscP could be a deleterious factor in our results.

A recent article from Di Sandro and colleagues [25] applies the comprehensive assessment of the transplantable tumor proposed by Mazzaferro [26] and included in the Italian Consensus-Based Approach to Organ Allocation in Liver Transplantation, concluding that high-risk patients, including partial response to bridging therapies or to downstaging, could benefit from a prioritization with a level of recurrence similar to intermediate-risk patients when transplanted quickly after re-staging. Even if they remain controversial (results of the article of Metha and Yao [27] propose a threshold to moderate the risk of selecting tumors with less favorable biology) and need wider validation, these results are interesting and could suggest, with our data, that the AFP model should be coupled to a prioritization system in order to improve the results of LT, including response to bridging therapies and downstaging.

It is interesting to see that a cholangiocarcinoma component on the explant analysis was an independent predictive factor for tumor recurrence (21% of recurrences during AFPscP and 7% during MilanCP). This underlines that graft allocation policy is only one factor influencing a patient’s prognosis; many others are involved and are changing over time.

Comparing MC and the AFP score, it is disturbing to observe the high risk of recurrence (22%) for patients exceeding the histological MC despite being within the AFP score in the AFPsc period. It is only 11.8% in the MilanCP, but significantly higher than the 4% of recurrence of patients within MC.

However, the radiological assessment of the AFP model instead of the histological one is of the best value because it was collected by a single and independent physician. According to these data, there is no difference in recurrence between patients outside and patients within MC, if they respected the AFP model; recurrence is around 7% in the MilanCP. Moreover, we observed again a slightly, though not significant, worse outcome in the AFPsc period (whether we focused on histological or radiological assessment); we think this is due to the period more than to the criteria. The compliance of physicians to the radiologic assessment of the AFPscore is excellent, 94% during the MilanCP and 96% during the AFPscP, better than it was for the MC in the MilanCP (86%). Radiologic assessment of the AFP score is a valuable tool, and according to it, the AFP score performed in prioritizing patients who would have been excluded by MC but who had acceptable outcomes; conversely, it identified patients within MC but who would have had worse outcomes (50% of recurrence).

A limitation of our study is that it was an observational, retrospective study, which is subject to calculation of the AFP score from imaging or histological reports without central revision. However, size and number of tumors as well as AFP every 3 months during the waiting time were prospectively recorded in the national registry. Another limitation is the absence of data on immunosuppressive treatment after liver transplantation. Based on the retrospective design of our study, it was considered too complex for a correct interpretation despite this factor being of interest for tumor recurrence. Because of the short period of time of this study, we can assume that immunosuppressive policy did not change greatly in each transplant center.

## 5. Conclusions

We found a similar agreement for the AFP model before and after its implementation, and we confirmed the value of AFP score for tumor recurrence prediction, but we observed a slightly worse prognosis of patients during the recent period, with several hypotheses (longer waiting time, more LRTs) but no clear explanation. The AFP model has recently been evaluated in Italy [28] and in Latin America [14], while the UK LT program discussed these criteria in the National Consensus Meeting. Our study emphasizes the influence of many parameters on recurrence: compliance of teams, criteria for downstaging, median waiting time, subtleties such as ‘temporary contraindication’ and maybe salvage transplantation. To assess the outcomes, prospective studies would be necessary for each different system.

## Figures and Tables

**Figure 1 cancers-13-02480-f001:**
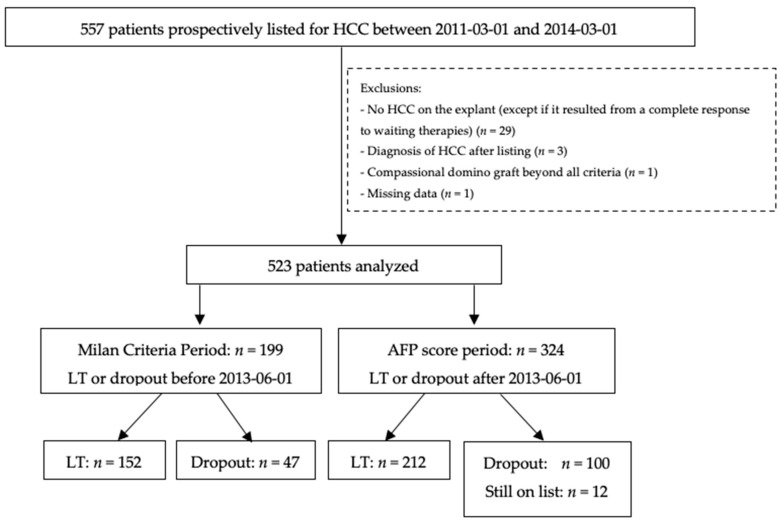
Study flow chart. HCC: Hepatocellular carcinoma; LT: Liver transplantation; AFP: α- fetoprotein

**Figure 2 cancers-13-02480-f002:**
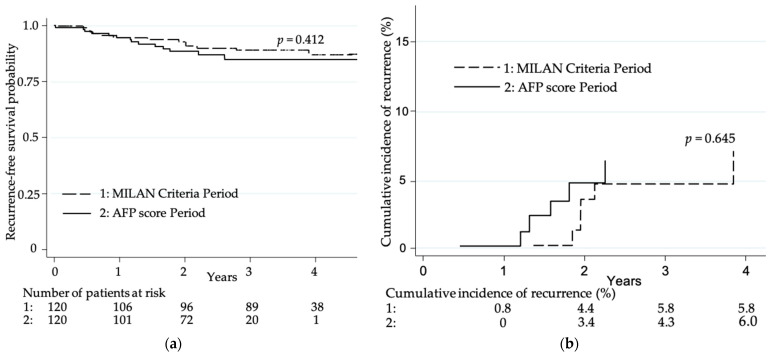
Risk of recurrence: (**a**) probabilities of recurrence-free survival according to Kaplan–Meyer estimates; (**b**) cumulative incidence of recurrence on competing analysis (Fine and Gray model).

**Figure 3 cancers-13-02480-f003:**
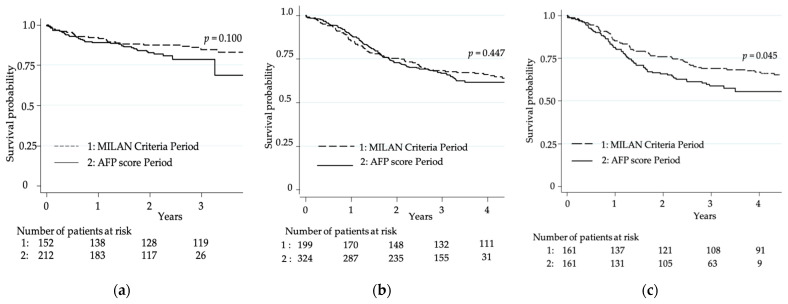
Overall survival. (**a**) Post-transplant overall probabilities of survival in MilanCP and AFPscP; (**b**,**c**) Post-listing overall probabilities of survival in MilanCP and AFPscP (**b**) without propensity matching and (**c**) with propensity score matching.

**Figure 4 cancers-13-02480-f004:**
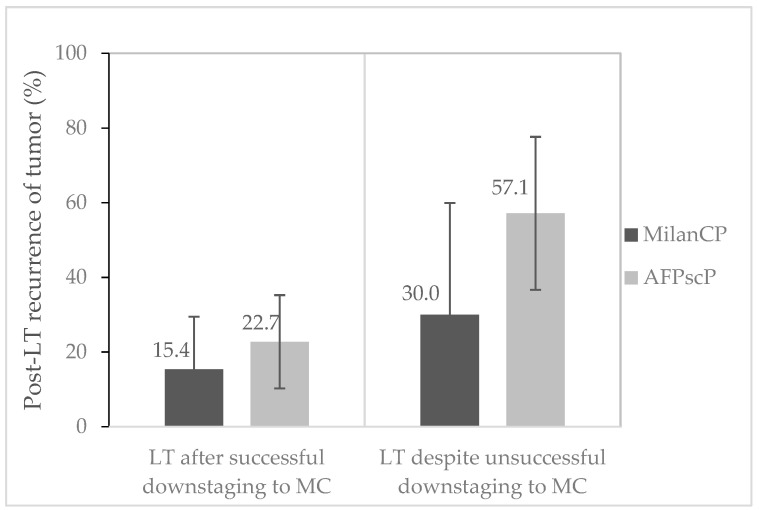
Post-transplant HCC (hepatocellular carcinoma) recurrence after a downstaging policy. Values are given as % of the corresponding population, with distinction of patients with successful downstaging (*n* = 26 in MilanCP, *n* = 44 in AFPscP) or not (*n* = 10 in MilanCP, *n* = 7 in AFPscP).

**Table 1 cancers-13-02480-t001:** Calculation of the AFP (α- fetoprotein) score. The score is calculated by adding the individual points for each obtained variable. A cut-off value of 2 separates between patients at high and low risk of recurrence [10].

Variables	β Coefficient	Hazard Ratio	Points
Largest diameter, cm			
≤3	0	1	0
3–6	0.272	1.31	1
>6	1.347	3.84	4
Number of nodules			
1–3	0	1	0
≥4	0.696	2.01	2
AFP level, ng/mL			
≤100	0	1	0
100–1000	0.668	1.95	2
>1000	0.945	2.57	3

**Table 2 cancers-13-02480-t002:** Comparative characteristics of the study population.

	MilanCP (*n* = 199)	AFPscP (*n* = 324)	*p* Value
Males (*n*, %)	179 (89.9%)	280 (86.4%)	0.232
Age at listing (years, median, (IQR))	58.7 (53.7–62.9)	59.4 (52.9–63.2)	0.639
Cirrhosis (*n*, %)	191 (96.0%)	309 (95.4%)	0.741
Causes of cirrhosis (*n*, %)			
Alcohol	78 (40.8%)	113 (36.6%)	0.340
Viral	50 (26.2%)	90 (29.1%)	0.476
Viral + Alcohol	23 (12%)	47 (15.2%)	0.321
NASH	8 (4.2%)	12 (3.9%)	0.866
BASH	20 (10.5%)	31 (10%)	0.875
Others	12 (6.3%)	16 (5.2%)	0.602
**Data at diagnosis**			
Pretreatment biopsy (n, %)	48 (24.1%)	80 (24.7%)	0.883
Number of tumors (median, (IQR))	1 (1–2)	1 (1–2)	0.985
Max diameter (mm, median, (IQR))	25 (20–35)	26 (20–38)	0.367
Sum of diameter (mm, median, (IQR))	34 (23–48)	34 (25–53)	0.750
AFP value, ng/mL (median, (IQR))	10 (5–24.8)	8.2 (5–21.3)	0.507
AFPsc diag: ≤2 vs. >2	176/23 (88.4% vs. 11.6%)	279/45 (86.1% vs. 13.9%)	0.441
**Data at listing**			
Pre-existing tumor treatment to avoid LT	52 (26.1%)	88 (27.2%)	0.796
Length between diagnosis and listing (days, median, (IQR))	270 (132–521)	238 (130–530)	0.402
Child–Pugh (*n*, %)			
A	91 (45.7%)	184 (56.8%)	**0.017**
B	60 (30.2%)	88 (27.2%)	0.591
C	48 (24.1%)	52 (16.1%)	**0.016**
MELD score (median, (IQR))	10.6 (8.2–15.7)	9.7 (7.6–14)	0.063
MELD > 20 (%, *n*)	26 (13.6)	25 (7.7)	**0.045**
Number of tumors (median, (IQR))	2 (1–3)	2 (1–3)	0.147
Largest diameter (median, (IQR))	24 (20–33)	26 (19–36)	0.428
Sum of diameter (mm, median, (IQR))	37 (24–54)	40 (26–61.5)	0.139
AFP value, ng/mL (median, (IQR))	8 (4.1–20.7)	7.4 (4–20)	0.583
Milan criteria: within vs. beyond	175/24 (87.9 vs. 12.1%)	278/46 (85.8 vs. 14.2%)	0.486
AFPsc-listing: ≤2 vs. >2	194/5 (97.5% vs. 2.5%)	307/17 (94.8% vs. 5.2%)	0.130
**Waiting time**			
Bridging treatments (*n*, %)	150 (75.4%)	273 (84.3%)	**0.012**
Number of treatments (median, (IQR))	2 (1–2) (1–5)	2 (1–3) (1–8)	**≤0.001**
Downstaging policy (*n*, %)	54 (34.8%)	92 (33.4%)	0.771
Unsuccessful downstaging (*n*, %)	28/54 (52%)	48/92 (50%)	0.971
Dropout of list (*n*, %, (IQR))	47 (23.6%) (17.9–30.1)	100 (30.9%) (25.9–36.2)	0.073
Dropout for HCC progression (*n*, %, (IQR))	35 (17.6%) (12.3–22.9)	67 (18.4%) (14.2–22.6)	0.926
Number of patients in ‘TCI’ (*n*, %)	68 (34%)	186 (57%)	**0.022**
Median ‘TCI’ time/patient (d, median, (IQR))	66 (41–153)	120 (54–374)	**≤0.001**
Number of patients still waiting	0	12	
**Last imaging before LT**	***N* = 152**	***N* = 212**	
Median time last imaging–LT (days, (IQR))	42 (18–74)	43.5 (21.5–67.5)	0.999
Number of tumors (median, (IQR))	2 (1–3)	2 (1–3)	0.340
Largest diameter (median, (IQR))	23.5 (18–31)	27 (20–35)	0.064
Sum of diameter (mm, median, (IQR))	40 (24–60)	45 (29–76)	0.085
AFP value, ng/mL (median, (IQR))	6 (3.6–19.9)	6 (3–13)	0.306
Last Milan criteria: within vs. beyond	130/21 (86.1% vs. 13.9%)	186/26 (87.7% vs. 12.3%)	0.646
Last-AFPsc: ≤2 vs. >2	142/9 (94% vs. 6%)	208/8 (96.2% vs. 3.8%)	0.331
**Transplantation data (*n* = 364)**	***N* = 152**	***N* = 212**	
Rate of liver transplantation	76.4%	69.1% *	**0.008**
Median waiting time (months) (IQR)	7.7 (3.7–12)	12.3 (8.2–16.1)	**<0.001**
LT consecutive to downstaging policy	36/152 (23.7%)	51/212 (24.1%)	0.949
Number of tumors (median, (IQR))	2 (1–4) (0–20)	2 (1–4) (0–50)	0.775
Largest diameter (mm, median, (IQR))	25 (15.5–35)	27.5 (17–37)	0.320
Sum of diameter (median, (IQR))	42 (27–67.5)	45 (29–77.5)	0.171
Milan criteria explant: within vs. beyond	100/52 (65.8% vs. 34.2%)	150/62 (70.8% vs. 29.2%)	0.314
AFPsc explant: ≤2 vs. >2	133/19 (87.5% vs. 12.5%)	187/25 (88.2% vs. 11.8%)	0.838
Cholangiocarcinoma component	8 (5.3%)	12 (5.7%)	0.802
Macrovascular invasion	13 (8.5%)	15 (7.1%)	0.527
Microvascular invasion	35 (23.0%)	50 (23.6%)	0.901
Satellites nodules	33 (21.7%)	43 (20.3%)	0.741
Major differentiation grade			
Not assessable	58 (38.2%)	79 (37.3%)	0.741
Well	44 (28.9%)	56 (26.4%)	0.593
Moderate	48 (31.6%)	72 (34%)	0.633
Poor	2 (1.3%)	5 (2.4%)	0.475
Percentage of tumor necrosis (median) Complete pathological response after bridging treatments	30.5 (0–82.5) 20/119 (16.8%)	36 (0–75) 25/179 (14.0%)	0.965 0.504
**Post-transplantation data (*n* = 364)**	***N* = 152**	***N* = 212**	
Follow-up (years, median, (IQR))	4.13 (1.81–4.81)	2.07 (1.49–2.65)	**<0.001**
Death (*n*, %)	26 (17.1%)	39 (18.4%)	0.751
Postoperative related death (*n*, %)	12 (8.5%)	18 (7.2%)	0.663
Tumor recurrence (*n*, % (IQR))	14 (9.2% (5.1; 15))	28 (13.2% (9; 18.5))	0.239

Abbreviations: * 12 patients still waiting. AFP: α- fetoprotein; BASH: Both Alcoholic and Steatotic Hepatitis; IQR: Interquartile range; LT: Liver transplantation; MELD: Model for End-stage Liver Disease; NASH: Non Alcoholic and non-alcoholic SteatoHepatitis; TCI: Temporary Contra-Indication.

**Table 3 cancers-13-02480-t003:** Risk factors for noncompliance of the histological AFP score: univariate and multivariate analyses (*n* = 364).

	Univariate Analysis			Multivariate Analysis		
Risk Factors	OR	95% CI	*p*	OR	95% CI	*p*
AFPscP/MilanCP	0.94	0.50–1.77	0.838	1.07	0.51–2.23	0.865
AFPsc diag > 2	1.99	0.92–4.34	0.082	NS		
Treatment to avoid transplantation	0.83	0.40–1.70	0.605			
Child–Pugh B	1.58	0.79–3.15	0.192			
MELD score > 20	0.46	0.11–2.00	0.300			
AFPsc listing ‘viable’ > 2	1.86	0.38–9.04	0.443			
Pre-LT bridging treatments	2.55	0.88–9.04	0.085	NS		
Number of bridging treatments	**1.43**	**1.09–1.86**	**0.009**	NS		
Downstaging policy	**4.38**	**2.28–8.41**	**<0.001**	**5.13**	**2.45–10.8**	**<0.001**
Waiting time > 14.5 months	1.69	0.86–3.32	0.128			
Beyond Milan on last imaging	**5.31**	**2.59–10.9**	**<0.001**	NS		
Last AFPsc ‘viable’ > 2	**33.0**	**10.2–108**	**<0.001**	**40.4**	**11.5–142**	**<0.001**

Abbreviations: LT: Liver transplantation; AFP: α- fetoprotein; MELD: Model for End-stage Liver Disease;

**Table 4 cancers-13-02480-t004:** Recurrence rate for each group of patients based on AFP score and Milan criteria on last-imaging assessment and on explant findings in transplanted patients during Milan criteria period of time and during AFP score period of time.

			Milan CP (*n* = 152)	AFPscP (*n* = 212)
Last imaging assessment	Within AFP score	Within Milan criteria	10/129 (7.8%)	22/186 (11.8%)
		Beyond Milan criteria	1/14 (7.1%)	2/18 (11.1%)
	Beyond AFP score	Within Milan criteria	1/2 (50%)	0
		Beyond Milan criteria	2/6 (33%)	4/8 (50%)
Explant assessment	Within AFP score	Within Milan criteria	4/100 (4%)	8/150 (5.3%)
		Beyond Milan criteria	4/34 (11.8%)	8/37 (22%)
	Beyond AFP score	Within Milan criteria	0	0
		Beyond Milan criteria	6/18 (33%)	12/25 (48%)

Abbreviations: AFP: α- fetoprotein.

**Table 5 cancers-13-02480-t005:** Risk factors for tumor recurrence: univariate and multivariate analyses (n = 364).

	Univariate Analysis			Multivariate Analysis		
Risk Factors	SHR	95% CI	*p*	SHR	95% CI	*p*
AFPscP/MilanCP	1.82	0.96–3.48	0.067	**2.34**	**1.16–4.73**	**0.017**
AFPsc diag > 2	**2.56**	**1.32–4.95**	**0.005**	NS		
αFP value diag > 100 ng/mL	1.74	0.74–4.09	0.202			
Treatment to avoid transplantation	**2.01**	**1.09–3.71**	**0.025**	**2.24**	**1.20–4.17**	**0.011**
Number of preemptive treatments	1.3	2.11	0.222			
Child B	0.90	0.45–1.80	0.760			
MELD > 20	0.69	0.22–2.15	0.520			
AFPsc listing > 2	1.87	0.50–7.00	0.354			
αFP value listing > 100 ng/mL	0.68	0.10–4.56	0.689			
Pre-LT bridging treatments	**3.43**	**1.08–10.87**	**0.036**	NS		
Number of bridging treatments	**1.37**	**1.13–1.65**	**0.001**			
Downstaging policy	**3.54**	**1.94–6.48**	**<0.001**	**2.50**	**1.30–4.81**	**0.006**
Waiting time	1.02	0.99–1.05	0.274			
Waiting time > 14.5 months	**2.02**	**1.06–3.84**	**0.032**	NS		
Last AFPsc > 2	**5.97**	**2.74–13.02**	**<0.001**	**2.58**	**1.06–6.30**	**0.038**
Last αFP value > 100 ng/mL	**5.26**	**2.59–10.68**	**<0.001**			
Macrovascular invasion	**5.98**	**3.07–11.67**	**<0.001**			
Microvascular invasion	**5.58**	**3.02–10.33**	**<0.001**	**2.61**	**1.17–5.81**	**0.018**
Satellites nodules	**2.59**	**1.38–4.84**	**0.003**			
Presence of intermediate differentiation	**3.13**	**1.10–8.94**	**0.033**			
Presence of poor differentiation	**7.55**	**2.28–25.0**	**0.001**			
Necrosis (for 10%)	0.97	0.91–1.05	0.472			
AFPsc explant > 2	**6.85**	**3.69–12.71**	**<0.001**	**2.82**	**1.14–6.99**	**0.025**
Cholangiocarcinoma component	**6.98**	**3.46–14.10**	**<0.001**	**5.22**	**2.65–10.30**	**≤0.001**

Abbreviation: AFP: α- fetoprotein; LT: Liver transplantation; MELD: Model for End-stage Liver Disease;

**Table 6 cancers-13-02480-t006:** Risk factors for post-transplantation death: univariate and multivariate analysis (n = 364).

	Univariate Analysis			Multivariate Analysis					
				If No Recurrence			If Recurrence		
Risk Factors	OR	95% CI	*p*	OR	95% CI	*p*	OR	95% CI	*p*
AFPscP/MilanCP	1.55	0.92–2.63	0.103	NS			1.95	1.00–8.17	0.051
AFPsc diag > 2	1.54	0.84–2.83	0.165						
αFP value diag > 100 ng/mL	1.44	0.69–3.02	0.336						
Treatment to avoid transplantation	**2.02**	**1.24–3.31**	**0.005**	1.93	0.99–3.64	0.054	NS		
Number of preemptive treatments	1.30	0.84–2.01	0.231						
Child B	**1.88**	**1.10–3.18**	**0.020**	**2.50**	**1.19–4.16**	**0.012**	1.77	0.85–4.47	0.077
MELD > 20	1.12	0.51–2.46	0.778						
AFPsc listing > 2	0.53	0.07–3.85	0.534						
αFP value listing > 100 ng/mL	1.46	0.46–4.66	0.520						
Pre-LT bridging treatments	1.11	0.59–2.09	0.737						
Number of bridging treatments	1.18	0.94–1.46	0.154						
Downstaging policy	**1.70**	**1.02–2.84**	**0.043**	NS			NS		
Waiting time	1.02	0.98–1.05	0.337						
Waiting time > 14.5 months	1.55	0.89–2.70	0.119						
Last AFPsc > 2	**3.06**	**1.46–6.43**	**0.003**	NS			NS		
Last αFP value > 100 ng/mL	1.75	0.84–3.67	0.138						
Macrovascular invasion	**2.75**	**1.47–5.15**	**0.002**	NS			NS		
Microvascular invasion	1.35	0.79–2.30	0.277						
Satellites nodules	1.13	0.649–2.02	0.669						
Presence of intermediate differentiation	1.30	0.67–2.51	0.443						
Presence of poor differentiation	**3.11**	**1.34–7.21**	**0.008**	NS			NS		
Necrosis	0.94	0.88–1.01	0.075						
AFPsc explant > 2	**3.44**	**2.01–5.87**	**<0.001**	NS			**2.84**	**1.52–9.74**	**0.005**
Cholangiocarcinoma component	**3.49**	**1.78–6.85**	**<0.001**	NS			NS		
Recurrence	**5.51**	**3.34–9.13**	**<0.001**						
Secondary other tumors	2.08	0.99–4.35	0.053						

Abbreviation: AFP: α- fetoprotein; LT: Liver transplantation; MELD: Model for End-stage Liver Disease;

## Data Availability

The data presented in this study are available on request from the corresponding author. They have been collected thanks to ABM reports and patient files in each medical center.

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
