# Peer review of "Liver Transplantation for Hepatocellular Carcinoma: A Real-Life Comparison of Milan Criteria and AFP Model"

_cancers, 2021, doi:10.3390/cancers13102480_

Round 1
Reviewer 1 Report
The clinical outcome of the liver transplantation for hepatocellular carcinoma in 5 French liver transplantation centers was analyzed, before and after the adoption of the AFP model. Although the overall survival rates were comparable between the two arms, post listing survival was better in the patients after the AFP model was adopted.
- (P1, Title) What is the intention to use “real-life”? Is it necessary?]
- (P1, L42 Abstract) It is described that “due to the increase in waiting time, of bridging therapies, downstaging policy and salvage transplantation”. It is impossible to know that the rate of the bridging therapies and downstaging policy in the AFPsc era from the data.
- (P3, patients and method section) What are the indication of the bridging therapies and downstaging policy? What are the changes of them from the Milan era to AFP model era?
- (P4, L170) What is 14.2@@@? Please clary it.
- (P5, Table 2) Are “Number of treatments” equal to “Number of curative treatments”? If, no, show them.
Author Response
Dear colleague,
Thank you for your comments and support concerning our article entitled "Liver transplantation for hepatocellular carcinoma: a real-life comparison of Milan criteria and AFP model".
Here are our responses, hoping it will appropriately meet your expectations.
1.(P1, Title) What is the intention to use “real-life”? Is it necessary?]
Real life refers to an absence of patient’s selection to enter into the study and to really study the effect of the change of allocation criteria. Indeed, all patients listed for HCC in the 5 participating centers were studied for this objective except very few in whom exclusion criteria were discovered later one.
The term is not mandatory. If the reviewer prefer to remove it, we totally agree.
2.(P1, L42 Abstract) It is described that “due to the increase in waiting time, of bridging therapies, downstaging policy and salvage transplantation”. It is impossible to know that the rate of the bridging therapies and downstaging policy in the AFPsc era from the data.
We thank reviewer 1 for this comment. Dowstaging was similar in both groups (34.8% and 33.4%, p=0.771). But downstaging policy is one of the independent predictive factor of tumor recurrence, as well as salvage transplantation. Similarly, the rate of bridging therapies was clearly different between groups (84% in the AFPsc vs 75% in the Milan CP, p=0.012).
Taking into account these points, our wording seems to be justified in the conclusion of the abstract and we did not modify it.
3.(P3, patients and method section) What are the indication of the bridging therapies and downstaging policy? What are the changes of them from the Milan era to AFP model era?
The bridging therapies and downstaging policy were proposed according to the decision of the multidisciplinary tumor board of each center. The bridging treatment were percutaneous ablation if unifocal tumor and trans-arterial chemo-embolization in case of multifocal tumor.
The indication of a downstaging policy was dedicated to the patients who were outside the criteria in use at the time of the tumor board meeting. In France it is not common to restrain the indication to the UCSF criteria as the upper limit of tumor burden and it is to each committee to decide if it is no longer reasonable to try a downstaging policy.
There was no change of bridging and downstaging policy between Milan era and AFP era in the considered participating centers.
Following your comment, we modified the Patients and Method section (P3 and P4, L136-139)
4.(P4, L170) What is 14.2@@@? Please clary it.
Thank you for your remark. This was a mistake, it has to be read Stata® software, version 14.2.
We corrected the sentence (P4 L170)
5.(P5, Table 2) Are “Number of treatments” equal to “Number of curative treatments”? If, no, show them.
In the Table 2, the ‘number of curative treatments’ is indicated in the section ‘Data at listing’ as ‘Pre-existing tumor treatment to avoid LT’, when patients received a curative treatment but a recurrence leading to their listing.
‘Number of treatments’ in the section ‘Waiting time’ refers to bridging therapies, without evident curative intentions. The number of surgeries or Radiofrequency Ablation used as bridging therapies is difficultly retrievable and led to recurrences except in the cases of the 12 patients still on list but temporarily inactivated.
Reviewer 2 Report
In their article entitled "Liver transplantation for hepatocellular carcinoma: a real-life comparison of Milan criteria and AFP model" Brusset and colleagues analysed the oncological outcomes of LT for HCC treatment, focusing on the prognostic accuracy of the AFP model and the Milan criteria and exploring the risk factors for tumor recurrence before and after the adoption of AFP model in the french allocation system
The subject of this research is interesting and study methodology appropriate; the results are appropriately discussed
I have few comments and requests to potentially improve this paper before its publication:
1) The Up-to-Seven (10.1016/S1470- 2045(08)70284-5) and the Metroticket 2.0 (10.1053/j.gastro.2017.09.025) cited in the introduction actually refer to two different models and publications - please correct (line 70)
2) please avoid comments on reported results (i.e. only - line 180 or line 195)
3) The response to neoadjuvant treatments (bridging and downstaging) should be reported according to mRECIST criteria; if such classification is difficultly retrievable, please include these lack of data in the study limitations
4) At line 229 the Authors stated that they have performed a Cox regression for identify the risk factors for laying beyond the AFP model on explant pathology: since the Cox regression is used to discriminate between multiple risk factors for the incidence of a defined event over time, I think that the Authors actually referred to a simple multivariate logistic regression (there is not a time-frame for being in or out the AFP model on explant pathology). If so, please correct appropriately.
5) Focusing again on explant pathology, it would be interesting to show the rate of complete pathological response after neoadjuvant treatments, especially considering the conclusions of the study that suggest a potentially increased risk of tumor recurrence after downstaging; if such data is difficultly retrievable, please include these lack of data in the study limitations
6) In the Methods section the Authors stated that "Successful Downstaging" referred to MC in the MC period and AFP score<2 in the AFP period (lines 139-140); so, why did they mention the MC while reporting the results of downstaging in the AFP period at line 313?
7) There is an error in the reference reported at line 406, it should refer to reference 24.
8) The Authors should better discuss the low performance of AFP model in discriminating between patients at high or low risk of recurrence after tumor downstaging.
If the longer waiting time and the higher rate of bridging treatments could explain the worse outcomes reported in AFP period, it must be pointed out that such policies are being widely applied in current clinical practice, increasing the need of further efforts in preoperative prediction of post-transplant outcomes.
Although the AFP model seems to be very useful for patient selection, its utility in prioritization is little discussed.
Considering the current patient-oriented allocation system that is applied in France, the suggestion of a further prioritization to LT after successfull downstaging could be helpful in improving the outcomes of this subclass of patients
I strongly suggest the Authors to refer to a paper that has recently been published on Cancers (doi: 10.3390/cancers11060741) that analyse this fundamental aspect of LT for HCC.
I congratulate to the Authors for their efforts and results
Best regards
Author Response
Dear colleague,
Thank you very much for your careful reading and comments concerning our article entitled "Liver transplantation for hepatocellular carcinoma: a real-life comparison of Milan criteria and AFP model".
Here are our responses and modifications following your comments, hoping it will appropriately meet your expectations.
1) The Up-to-Seven (10.1016/S1470- 2045(08)70284-5) and the Metroticket 2.0 (10.1053/j.gastro.2017.09.025) cited in the introduction actually refer to two different models and publications - please correct (line 70)
Thank you for your remark, the sentence was unclear as it referred only to the revised Up-to-Seven criteria in 2018. A correction is made to mention the first publication of 2009 also, L70.
2) please avoid comments on reported results (i.e. only - line 180 or line 195)
Thank you for your remark, removals have been made (now corresponding line 185 and line 207).
3) The response to neoadjuvant treatments (bridging and downstaging) should be reported according to mRECIST criteria; if such classification is difficultly retrievable, please include these lack of data in the study limitations
The response to neoadjuvant treatments has been added in the text (L211-217), in order not to overload Table 2:
In the MilanCP, 274 bridging treatments were realized: complete response (CR) was obtained after 97 treatments (35.4%), partial response (PR) after 119 treatments (43.4%), stable disease (SD) after 17 treatments (6.2%) and progressive disease (PD) after 27 treatments (9.8%). In the AFPscP, 621 bridging treatments were realized: CR was obtained after 219 treatments (35.3%), PR after 266 treatments (42.8%), SD after 43 treatments (6.9%) and PD after 73 treatments (11.7%). The proportions were not different between the two periods (p=0.633).
4) At line 229 the Authors stated that they have performed a Cox regression for identify the risk factors for laying beyond the AFP model on explant pathology: since the Cox regression is used to discriminate between multiple risk factors for the incidence of a defined event over time, I think that the Authors actually referred to a simple multivariate logistic regression (there is not a time-frame for being in or out the AFP model on explant pathology). If so, please correct appropriately.
Thank you for your remark on this mistake, the modification has been made (now corresponding line 253).
5) Focusing again on explant pathology, it would be interesting to show the rate of complete pathological response after neoadjuvant treatments, especially considering the conclusions of the study that suggest a potentially increased risk of tumor recurrence after downstaging; if such data is difficultly retrievable, please include these lack of data in the study limitations
We completed the Table 2 with the following data: the rate of complete pathological response for patients undergoing bridging therapies (included or not in a downstaging policy) was not different in the two subgroups with 16.8% in MilanCP vs 14% in AFPscP, p=0.504.
In the subgroup “downstaging policy”, 3 patients on the 36 transplanted during the MilanCP had a complete pathological response (8.3%), vs 6 patients on the 51 transplanted during the AFPscP had a complete pathological response (11.7%), without significative difference (p=0.730).
Inside the MilanCP, there was a trend of more complete pathological response if there was not a downstaging policy (20.5% vs 8.3%, OR 2.81 IC95% [0.73-16.0], p=0.118). Inside the AFPscP, there was no difference of complete pathological response according to a downstaging policy (14.8% vs 11.7%, OR 1.3 IC95% [0.46-4.26], p=0.811).
These data are added on the paragraph on downstaging, L341-343, L354-359.
6) In the Methods section the Authors stated that "Successful Downstaging" referred to MC in the MC period and AFP score<2 in the AFP period (lines 139-140); so, why did they mention the MC while reporting the results of downstaging in the AFP period at line 313?
Thank you for your careful reading. We corrected this mistake. A single definition of “reintegration of the Milan Criteria” was used for the definition of a successful downstaging.
We modified the sentence lines 149-150, and a mistake in Table 2 accordingly.
7) There is an error in the reference reported at line 406, it should refer to reference 24.
Reorganization of references have been made.
8) The Authors should better discuss the low performance of AFP model in discriminating between patients at high or low risk of recurrence after tumor downstaging.
If the longer waiting time and the higher rate of bridging treatments could explain the worse outcomes reported in AFP period, it must be pointed out that such policies are being widely applied in current clinical practice, increasing the need of further efforts in preoperative prediction of post-transplant outcomes.
Although the AFP model seems to be very useful for patient selection, its utility in prioritization is little discussed.
Considering the current patient-oriented allocation system that is applied in France, the suggestion of a further prioritization to LT after successfull downstaging could be helpful in improving the outcomes of this subclass of patients I strongly suggest the Authors to refer to a paper that has recently been published on Cancers (doi: 10.3390/cancers11060741) that analyse this fundamental aspect of LT for HCC.
We thank reviewer 2 for this last comment. Indeed, AFP score has a poor discriminating power for high and low risk of tumor recurrence after downstaging. To further highlighted this point, we added a comment with the mentioned paper in the discussion chapter (L409-427).
Round 2
Reviewer 2 Report
I congratulate with the Authors for their further efforts
The corrections they made (inclusion o mRECIST response along with some data addition, minor statistical corrections, and further discussion on the performance of AFP model on downstaged tumors) significantly improved the quality of the paper, that in my opinion is now totally suitable for publication
Best regards